# Standalone Flow Diversion Therapy Effectively Controls Rebleeding of Acutely Ruptured Internal Carotid Artery Trunk (Nonbranching) Microaneurysms

**DOI:** 10.3390/jcm10225249

**Published:** 2021-11-11

**Authors:** José E. Cohen, Hans Henkes, John Moshe Gomori, Gustavo Rajz, Ronen Leker

**Affiliations:** 1Department of Neurosurgery, Hadassah-Hebrew University Medical Center, Jerusalem 91120, Israel; 2Neuroradiologische Klinik, Klinikum Stuttgart, 70174 Stuttgart, Germany; hhhenkes@aol.com; 3Department of Radiology, Hadassah-Hebrew University Medical Center, Jerusalem 91120, Israel; gomori@md.huji.ac.il; 4Department of Neurosurgery, Shaare Zedek Medical Center, Jerusalem 91031, Israel; rajzgustavo@hotmail.com; 5Department of Neurology, Hadassah-Hebrew University Medical Center, Jerusalem 91120, Israel; leker@hadassah.org.il

**Keywords:** blister aneurysm, brain aneurysm, flow diverter stent, hemorrhagic stroke, internal carotid artery, microaneurysm, subarachnoid hemorrhage

## Abstract

Flow diversion is a promising option in selected patients with acutely ruptured microaneurysms. In this article, we reviewed our experience. Patients with acute spontaneous subarachnoid hemorrhage (SAH) after rupture of a blister-like or saccular microaneurysm (≤2 mm maximal diameter) at a nonbranching ICA site treated from January 2016 to June 2019 using flow diversion as standalone therapy were included in this study. An EVD was usually placed preventively. Antiplatelet effects of pre-procedure DAPT were evaluated (target PRU, 80–160). After the intervention, DAPT was continued for ≥6 months, aspirin—indefinitely. Angiographic controls were obtained. Fifteen patients (12 female; mean age, 46.4 years) with 15 ruptured ICA microaneurysms (mean diameter, 1.8 mm) were included. An EVD was placed in 12 patients (75%) before DAPT administration and stenting. PRU values immediately before FDS were 1–134 (mean, 72.1). One patient died 27 days after flow diversion due to a suspected fulminant pulmonary embolism. Aneurysms were completely occluded at the 6–12-month angiographic follow-up in 14/14 surviving patients, with no rebleeding at a mean of 14 months. Late mRS was 0–2 in 13/14 patients and 3 in one due to sequelae of the original hemorrhage. Flow diversion provided robust aneurysm rebleeding control. Angiographic follow-up confirmed complete aneurysm occlusion in all the cases.

## 1. Introduction

Aneurysms located at nonbranching sites in the supraclinoid internal carotid artery (ICA) are rare, comprising only 0.9–6.5% of all ICA aneurysms. Usually, they are small-sized and broad-based. They may protrude from the dorsal wall of the ICA [1] or present as blood blister-like aneurysms of the superior [2] or anterior ICA wall [3], or more generally as ICA trunk aneurysms [4]. ICA trunk aneurysms can be divided into blister- and saccular-type lesions. They differ in shape, wall histology, and treatment requirements [4,5]. Blister-type aneurysms arise more frequently from the anteromedial wall than the posterolateral aspect of the supraclinoid ICA [4] while saccular aneurysms are prevalent in the latter. However, it is frequently difficult to differentiate between these subtypes since angiographic evidence of a saccular shape does not always correlate with the nature of the aneurysm wall and overlapping features and progression have been described [4,6]. Blister aneurysms generally present a more fragile wall than saccular aneurysms [2,5], are unpredictable clinically and during surgery, and generally have a poor prognosis. Management is challenging even in expert hands. They are associated with higher rates of intraoperative rupture and greater morbidity compared with saccular aneurysms [7].

Endovascular flow diversion techniques are generally considered inadequate for the management of ruptured aneurysms because of the intrinsic inability of stents to achieve immediate complete aneurysm occlusion as well as concern over the need for double antiplatelet therapy (DAPT). However, flow diversion has recently emerged as a promising option for the treatment of blister aneurysms of the supraclinoid ICA, with studies showing low rates of repeat hemorrhage and procedural complications and a high rate of long-term occlusion [8,9].

In this study, we review our experience with flow diversion for ruptured microaneurysms of the nonbranching ICA trunk using a protocol based on early ventriculostomy and antiplatelet therapy targeting a low level of P2Y12 reaction units (PRU).

## 2. Materials and Methods

### 2.1. Patients

Patients presenting with acute spontaneous subarachnoid hemorrhage (SAH) secondary to rupture of a microaneurysm (≤2 mm maximal diameter) at a nonbranching ICA site including both blister-like aneurysms (shallow broad-based on the supraclinoid side wall) and saccular microaneurysms treated from January 2016 to June 2019 using flow diverter stent techniques as standalone therapy were included in this study. Patients with other types of aneurysms, those treated with coiling techniques with or without stenting, and patients presenting with an absolute contraindication for antiplatelet therapy were excluded.

Demographic, clinical, angiographic, procedural data, and clinical and radiological outcomes were collected and analyzed through the latest available follow-up. SAH was confirmed by CT imaging and CTA was obtained in every case to evaluate intracranial and extracranial vessels. Hunt–Hess score as well as Fisher grade were recorded.

Approval for prospective data collection and retrospective analysis of all interventional procedures reported in this study was given by the institutional review board of the medical faculty. Informed consent for study inclusion was obtained from the patients or their legal representatives.

### 2.2. Criteria for External Ventricular Drain Placement

The need for an external ventricular drain (EVD) was assessed based on clinical indications, mainly altered consciousness and neuroradiological data, including ventricular dilatation, hemorrhage size, and risk of hydrocephalus. In acute cases where stent implant was anticipated, an EVD was preventively placed in most patients, with the exception of those with unaltered sensorium or minor hemorrhage (Fisher grades 1–2). This strategy is based on evidence that EVD placement in patients on DAPT is associated with increased risk of hemorrhagic complications [10,11]. If permanent shunt placement or any other interventional procedure was required after FDS implantation, the double antiplatelet plan was transiently modified so that the procedure could be carried out under aspirin alone. Ventricular catheter shunts were replaced via the same tract and burr hole as those used for the original EVD.

### 2.3. Antiplatelet Strategy

In cases when an FDS implant was anticipated, the patients were premedicated with aspirin and clopidogrel (300–500 mg and 300–600 mg, respectively), PO or per nasogastric tube. Platelet function assays revealed a measurable antiplatelet effect starting within 60 min and reaching a maximum of antiplatelet inhibition 2 h after the 600 mg loading dose of clopidogrel. In our practice, as a general rule before June 2017, the patients who were treated on admission day were loaded with 600 mg of clopidogrel and its effects were evaluated after 2 h. When treatment was planned for on the next day, the patients received only a 300 mg loading dose and clopidogrel effects were evaluated before the intervention, usually 6–18 h after drug administration. The antiplatelet effects of clopidogrel were evaluated with the Verify Now Platelet Reactivity Test (Accriva Diagnostics, San Diego, CA, USA), targeting a PRU of 80–160. The patients who were hypo-responders (PRU > 200) or non-responders to clopidogrel were managed with either ticagrelor (loading dose, 180–270 mg; postoperative maintenance, 60–90 mg/12 h) or prasugrel (loading dose, 40–60 mg; postoperative maintenance, 5–10 mg/day). Platelet function was checked again in all the patients on any of the drugs 2 h after administration of the loading dose, prior to the intervention. In general, the hyper-responders to clopidogrel were prescribed lower maintenance doses (usually, 75 mg every 48 h), and aspirin was reduced (75 or 81 mg/day or alternating days) or discontinued.

Due to the delayed and unpredictable effects of clopidogrel, prasugrel or ticagrelor were used instead of clopidogrel beginning in June 2017. This strategy reduced the interval between antiplatelet medication administration and confirmation the PRU target had been reached, enabling the procedure to commence.

In cases where an FDS implant was not anticipated and the patients were therefore not previously loaded with aspirin and clopidogrel, a single dose of aspirin (300 mg) and prasugrel or ticagrelor (30–50 mg or 180 mg, respectively) were administered via nasogastric tube and antiplatelet effects were evaluated when the decision is made to move forward with treatment after diagnostic angiography. After confirming a strong antiplatelet effect (PRU < 120), the patients received a single bolus of 3000–4000 units of heparin to reach the target international normalized ratio (INR) of 250–300 s. Following the procedure, DAPT was indicated for a minimum period of 6 months, and aspirin was indicated indefinitely.

### 2.4. Neuroendovascular Technique

Treatment decisions were made by a consensus of vascular and endovascular neurosurgeons. Every procedure was performed via transfemoral approach under general anesthesia. After gaining arterial access, a diagnostic angiogram was obtained and the target aneurysm and parent vessel were analyzed and measured. The target vessels were evaluated with a biplane angiography system and 3D rotational angiography.

A 6F guiding sheath (Arrowsheath, Arrow, Kington, UK) was then placed at the origin of the ICA after diagnostic catheter exchange maneuvers. A Navien 0.058″ catheter (Medtronic, Minneapolis, MN, USA) was used as an intermediate supporting catheter in all the procedures. FDS length was chosen according to the length of the aneurysm neck and based on a procedural goal of ensuring arterial wall coverage with the inner mesh extending at least 5 mm beyond the distal and proximal limits of the neck. For FDS delivery, an Exelsior XT-27 microcatheter (Stryker, Kalamazoo, MI, USA) or a Headway 27 (Microvention Aliso, Viejo, CA, USA) was navigated past the aneurysm neck with the assistance of Synchro (Stryker) or pORTAL (phenox) microguidewires. Under roadmap guidance, the FDS (PED shield, Medtronic; or p64, phenox, Bloomberg, Germany) was then deployed by withdrawing the delivery microcatheter and pushing the delivery wire. If incomplete stent opening or suboptimal wall apposition was observed on radioscopy or control angiography, stent angioplasty was performed with the aid of compliant balloons (HyperGlide or HyperForm, Covidien, Irvine, CA, USA; or Eclipse 2L, Balt Extrusion, Montmorency, France).

Angiographic controls were obtained after 3–6 months and 12 months. Further angiographic controls were performed only in cases of incomplete aneurysm exclusion.

## 3. Results

### 3.1. Patient Characteristics

From January 2016 to June 2019, we treated 15 patients with 15 ruptured ICA microaneurysms who met study criteria (12 females [80%], mean age 46.4 years [range 37–72]). Patient and procedural details are summarized in Table 1. Nine aneurysms were located on the right intra dural nonbranching ICA, and six on the left. Three patients presented a second aneurysm that was not considered to be the source of hemorrhage. The mean aneurysm size was 1.8 mm (range, 0.4–2.0 mm). Based on topography and angioarchitecture, eight aneurysms were defined as blister (Figure 1), seven—as saccular (Figure 2); however, one of the blister aneurysms (case No. 6) evolved to saccular topography.

All 15 patients presented with spontaneous SAH detected on admission CT. Admission Hunt–Hess grade ranged from 1 to 5 (mean 2.8), and Fisher score ranged from 2 to 4 (mean 3.1). An EVD was placed in 12 patients (75%) before administration of DAPT and before the endovascular intervention; in nine because of a combination of neurological status and hemorrhage severity, and in three borderline cases in anticipation of the need for DAPT and an FDS implant. No patient required post-intervention EVD placement due to hydrocephalus.

The endovascular procedure was performed on SAH day 1–8 (average 2.5). Two antiplatelet agents were administered in all the patients: aspirin–prasugrel in 10 patients, aspirin–clopidogrel in four patients, and aspirin–ticagrelor in one. Periprocedural PRU values obtained immediately before FDS implant ranged from 1 to 134 (mean 72.1).

### 3.2. Treatment Details and Adverse Events

All the aneurysms were managed with FDS as monotherapy. Every patient was treated with a single device (PED shield in 13 cases and p64 in two) and balloon angioplasty was required in only two cases. No procedural complications occurred. Immediately after FDS implantation, aneurysms presented different degrees of incomplete angiographic occlusion, with O’Kelly–Marotta scores [12] of A1 in three patients, B1 in four, B2 in one, C2 in six, C3 in one (Table 1). Despite consistent near-zero or incomplete angiographic exclusion, none of the aneurysms rebled during the hospital stay or follow-up. One patient (No. 9) developed a femoral pseudoaneurysm that was managed with percutaneous injection of thrombin. Two patients complicated with pneumonia. One patient (No. 5) developed symptomatic cerebral vasospasm that was refractory to conservative measures and required endovascular angioplasty. One patient (No. 6) died 27 days after the endovascular intervention and 3 days after transfer to rehabilitation after a suspected fulminant pulmonary embolism.

### 3.3. Follow-Up Evaluation

Angiographic follow-up was available in all the 14 surviving patients at a mean of 6 months (range 5–8 months) post-treatment and demonstrated complete occlusion in 14/14 aneurysms.

The mean duration of clinical follow-up was 14 months (range 6–24 months) after the intervention. There was a favorable outcome (modified Rankin score (mRS), 0–2) in 13/14 surviving patients (mRS 0 in four patients, 1 in five, 2 in four). One patient (No. 14) had a 90-day mRS of 3 due to sequelae of the original hemorrhage.

## 4. Discussion

In this single-center case series, we report our experience in the endovascular treatment of acutely ruptured microaneurysms of the nonbranching ICA vasculature by means of a reconstructive FDS monotherapy technique. This approach was feasible in all cases with the implantation of a single device. There were no procedural ischemic or hemorrhagic complications, and flow diversion provided robust aneurysm rebleeding control despite consistent non-exclusion of the aneurysm on immediate angiographic controls. Angiographic follow-up after approximately 6 months confirmed a complete aneurysm occlusion rate of 100%. Our results compare favorably to other endovascular and surgical approaches in terms of procedural complications, rebleeding rates, and rates of complete aneurysm occlusion at medium-term follow-up.

The need for DAPT after FDS implantation is an important concern for surgeons; however, in a recent comprehensive meta-analysis of the literature, morbidity and mortality associated with blood blister aneurysms was lower than reported rates with surgical treatment even in the presence of DAPT [13]. The policy of low-threshold early ventriculostomy placement reduced our concern regarding the risk of hemorrhagic complications following urgent ventriculostomy in patients on DAPT. No patient in this series required placement of ventriculostomy after DAPT was initiated.

Our protocol included routine antiplatelet effect testing, FDS implantation under a low PRU (target > 120), and routine intraprocedural heparinization (activated clotting time (ACT), 250–300). This practice resulted in a low rate of intraprocedural and periprocedural thromboembolic complications compared with other series that also used PED in the majority of the cases, and remarkably this strategy was not associated with procedural hemorrhagic complications. Mokin et al. reported a multicenter series of 43 consecutive cases of blood blister aneurysms in the supraclinoid ICA treated with the PED FDS [9]. There were five procedural thromboembolic complications in the series (11.6%) and two hemorrhagic complications (4.6%). Technical considerations such as administration and dosing of antiplatelets and post-procedure care were left to the preference of each of the interventionalists.

Rebleeding protection in the immediate post-procedure period has been attributed to the reduction of a jet inflow of blood, resulting in lowered hemodynamic sheering stress on the aneurysmal wall. This alleviates the most significant factors that predispose these aneurysms to re-rupture and rebleeding [14]. This protective effect might be more robust for microaneurysms in comparison to larger aneurysms. A recent meta-analysis of 20 studies evaluating 223 patients with acutely ruptured intracranial aneurysms treated with FDS reported that 4% of aneurysms rebled [15]; however, rebleeding has been seen infrequently after treating blister or small aneurysms.

Chalouhi et al. reported their experience with five patients who presented with ICA blister aneurysms that were treated with FDS [16]. They reported no procedural complications or rebleeding, and their patients experienced good clinical outcomes with complete aneurysm occlusion in all the cases. Lyn et al. reported similar findings after treating seven patients, with no procedural complications, no rebleeding, excellent clinical evolution, and complete occlusion in all the cases [17]. Linfante et al. treated eight patients and reported no procedural complications, with 100% occlusion at a mean 7-month follow-up [8]. Mokin reported one case of intraprocedural aneurysm bleeding and one case of aneurysm rebleeding 4 days after FDS implantation in 43 patients treated with the PED. This group reported complete aneurysm occlusion rates of 87.5% at a mean of 4 months after FDS implantation [9]. These reports correspond well with our results. We can speculate that the negligible rebleeding rates observed in these series could be partly related to the small size that is characteristic of most ICA trunk aneurysms.

The nosological categorization between blister and saccular microaneurysms based on angiographic criteria is not precise; however, there seems to be no difference between these subgroups in procedural complication or occlusion rates after FDS implantation. For ICA trunk aneurysms, this differentiation is more relevant for surgical procedures than for FDS-based endovascular interventions.

## 5. Conclusions

In this series of consecutive patients, flow diversion was a safe and effective treatment for this subset of acutely ruptured aneurysms. A low threshold for pre-procedure EVD placement and antiplatelet effectiveness monitoring are key factors in lessening the frequency of hemorrhagic and thromboembolic complications.

## Figures and Tables

**Figure 1 jcm-10-05249-f001:**
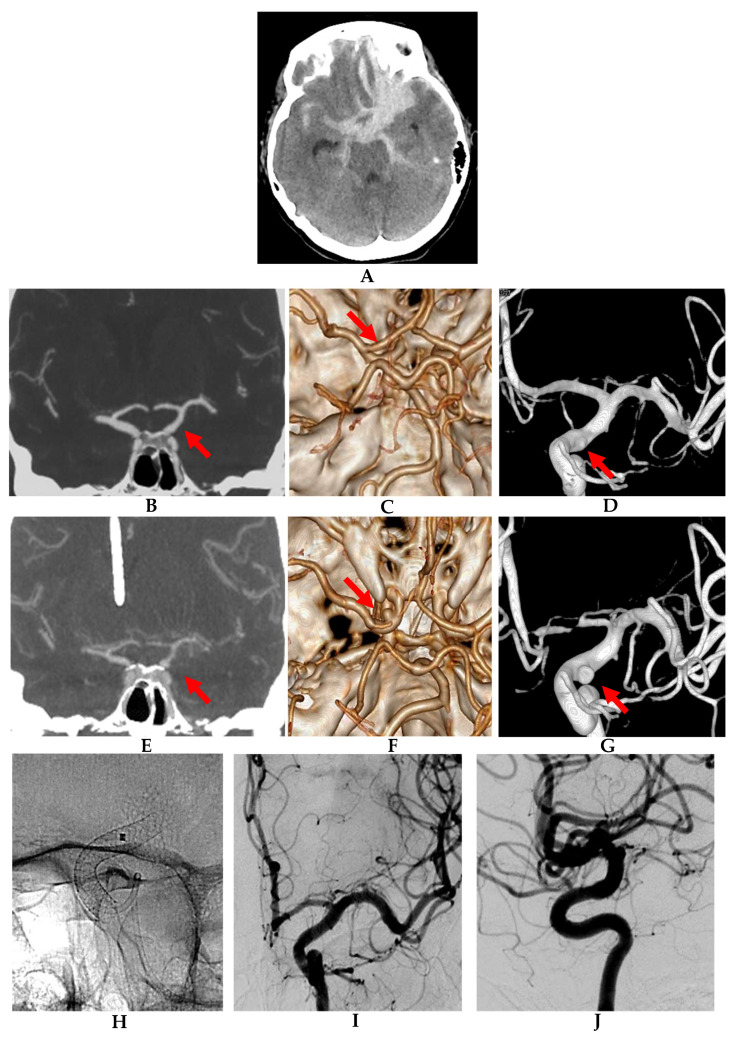
Growing blister aneurysm in a 55-year-old female (patient 5). (**A**) Admission non-contrast head CT showing extensive basal subarachnoid hemorrhage (SAH) lateralized to the left carotid cisterns and extending subfrontally. (**B**) Admission coronal CT angiogram (CTA), (**C**) CTA reconstruction image, and (**D**) 3D reconstruction of a digital subtraction angiogram on anteroposterior view showing signs of subtle vasculopathy at the ventrolateral wall of supraclinoid left ICA (red arrow), but not a clear aneurysm. (**E**) Coronal CTA, (**F**) CTA reconstruction image, and (**G**) 3D reconstruction of a digital subtraction angiogram on anteroposterior view obtained on day 8 after subarachnoid hemorrhage (SAH) showing development of angiographic vasospasm and clearly depicting a left ICA supraclinoid aneurysm (red arrows). (**H**) Magnified radioscopic image of the implanted flow diverter stent. (**I**,**J**) Digital subtraction angiograms of the left ICA obtained 4 months after the endovascular intervention, anteroposterior and lateral magnified views, showing remodeling of the stented arterial segment with no signs of residual aneurysm and only mild signs of in-stent stenosis.

**Figure 2 jcm-10-05249-f002:**
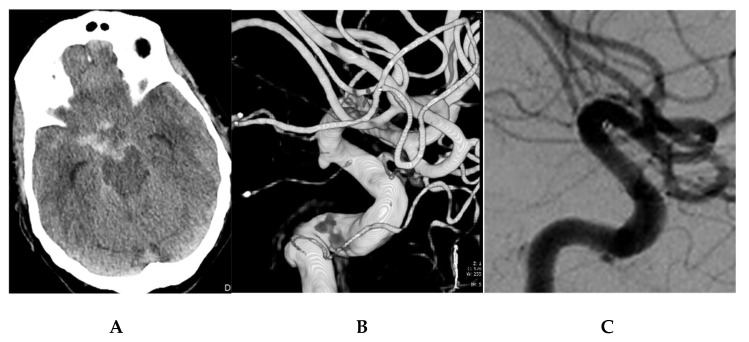
Saccular aneurysm in a 67-year-old female (patient 12). (**A**) Admission CT showing a SAH, Fisher grade 3, centered on the right carotid cistern. (**B**) 3D reconstruction of a rotational angiogram showing a saccular microaneurysm at a nonbranching point of the posterior wall of the supraclinoid right ICA, posterior communicating artery segment. The aneurysm was treated by implantation of a PED 4 × 16 mm FDS on SAH day 3. (**C**) Angiographic follow-up obtained after 5 months showing complete exclusion of the treated aneurysm and no sign of in-stent stenosis.

**Table 1 jcm-10-05249-t001:** Patient characteristics, presentation, procedural details, complications, and outcomes.

Patient No. (Age/Sex)	Hunt–Hess Grade	Fisher Grade	EVD	Aneurysm Location	DAPT Plan	Pre-Procedure PRU	SAH Day	FDS Type, Size	Procedural Complications	O’Kelly–Marotta Procedure/6-mo. Follow-Up	mRS	
											90 Days	180 Days
1–47/F	2	4	Yes	R-ICA Anterior wall	A + P	122	2	PEDs 3.5 × 18	No	B1/D	2	1
2–64/F	3	3	Yes	R-ICA Anterior wall	A + P	82	3	PEDs 3.5 × 18	No	B2/D	1	1
3–44/F	2	2	No	L-ICA Anterior wall	A + C	134	1	PEDs 3.75 × 18	No	A1/D	2	2
4–63/M	4	4	Yes	L-ICA Anterior wall	A + C	77	2	PEDs 4.25 × 16	No	C2/D	3	3
5–55/F (Figure 1)	2	3	Yes	L-ICA Anterolateral paraoph	A + C	54	8	PEDs 4.0 × 20	No	C3/D	1	0
6–72/F	3	3	Yes	R-ICA Lateral paraoph	A + P	8	2	PEDs 3.75 × 18	No	A1/-	6	6
7–56/F	2	3	Yes	L-ICA Medial paraoph	A + P	18	2	PEDs 3.25 × 16	No	B1/D	2	1
8–61/F	5	4	Yes	R-ICA Medial paraopth	A + P	114	1	PEDs 3.50 × 18	No	B1/D	3	3
9–42/F	4	4	Yes	R-ICA PcomA sg	A + P	1	2	PEDs 3.75 × 18	Femoral PSA	C2/D	2	2
10–62/M	2	2	No	L-ICA PcomA sg	A + T	65	1	p-6 44.0 × 18	No	A1/D	1	0
11–37/F	2	3	Yes	R-ICA PcomA sg	A + P	86	2	PEDs 3.75 × 18	No	B1/D	1	0
12–67/F (Figure 2)	3	3	Yes	R-ICA PcomA sg	A + P	68	3	PEDs 4.00 × 16	No	C2/D	1	1
13–54/F	3	2	Yes	L-ICA PcomA sg	A + P	101	2	PEDs 3.25 × 16	No	C2/D	2	2
14–39/F	4	4	Yes	R-ICA Acha sg	A + C	132	4	PEDs 2.5 × 16	No	C2/D	3	3
15–42/M	1	2	No	R-ICA Acha sg	A + P	19	3	p-6 44.0 × 16	No	C2/D	2	0

A—aspirin; C—clopidogrel; P—prasugrel; T—ticagrelor; L—left; R—right; AchA—anterior choroid artery; DAPT—dual antiplatelet therapy; EVD—external ventricular drain; FDS—flow diverter stent; ICA—internal carotid artery; mRS—modified Rankin score; paraoph—paraophthalmic artery; PcomA—posterior communicating artery; PRU—P2Y12 reaction units; PSA—pseudoaneurysm; SAH—subarachnoid hemorrhage; Sg—segment.

## Data Availability

The data presented in this study are available on written request to the corresponding author. The data are not publicly available to protect patient confidentiality.

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
