# Peer review of "Standalone Flow Diversion Therapy Effectively Controls Rebleeding of Acutely Ruptured Internal Carotid Artery Trunk (Nonbranching) Microaneurysms"

_jcm, 2021, doi:10.3390/jcm10225249_

Round 1
Reviewer 1 Report
Although it is an experience with the use of flow diverters in the treatment of subarachnoid hemorrhage caused by ruptured blister-like aneurysm, as the authors cite, the usefulness of this treatment has already been reported in a larger group of patients.
The use of flow diverters requires firm efficacy of antiplatelet agents, and antithrombotic therapy is at odds with the treatment of subarachnoid hemorrhage, a hemorrhagic stroke. It is of interest to investigate the use of effective antiplatelet agents using PRU, but I think that more exploration of this may be of greater interest to the reader.
Clopidogrel is a prodrug that is said to be required for about 24 hours until the onset of efficacy. In this regard, I would like to know how many doses of each drug were taken until each patient achieved the PRU target and how many PRU measurements were taken. It has also been reported that bleeding complications increase when the PRU is 60 or less, so it is considered useful to indicate a protocol for adjusting the dose of clopidogrel after flow-diverter placement and repeat the PRU measurement. Please consider.
Author Response
Although it is an experience with the use of flow diverters in the treatment of subarachnoid hemorrhage caused by ruptured blister-like aneurysm, as the authors cite, the usefulness of this treatment has already been reported in a larger group of patients.
Response: In these series, there the strategy for managing antiplatelet meds is not discussed. This is the key to our low rate of hemorrhagic complications and we hope discussion of our protocol provides a useful addition to the literature. As suggested by the reviewer, we have added a information about the protocol to the Discussion section.
The use of flow diverters requires firm efficacy of antiplatelet agents, and antithrombotic therapy is at odds with the treatment of subarachnoid hemorrhage, a hemorrhagic stroke. It is of interest to investigate the use of effective antiplatelet agents using PRU, but I think that more exploration of this may be of greater interest to the reader.
Response: We have added comments on the use of PRU to monitor antiplatelet effects to the Discussion.
Clopidogrel is a prodrug that is said to be required for about 24 hours until the onset of efficacy. In this regard, I would like to know how many doses of each drug were taken until each patient achieved the PRU target and how many PRU measurements were taken. It has also been reported that bleeding complications increase when the PRU is 60 or less, so it is considered useful to indicate a protocol for adjusting the dose of clopidogrel after flow-diverter placement and repeat the PRU measurement. Please consider.
Response: In our experience, platelet function assays revealed a measurable antiplatelet effect starting within 60 minutes and reaching a maximum of antiplatelet inhibition 2 hours after the 600-mg loading dose of clopidogrel (1-3). In our practice, as a general rule, before June 2017, patients who were treated on admission day were loaded with 600 mg of clopidogrel and its effects were evaluated after 2 hours. When treatment was planned for on next day, patients received only a 300 mg loading dose and clopidogrel effects were evaluated before the intervention, usually in the 6-18 hours period after drug administration. PRU <60 indicates a hyper-response, hence our target range is 80-160.
We have added more detail on management of these medications in Methods, in the section focusing on antiplatelet management.
- Fefer, P.; Hod, H.; Hammerman, H.; Segev, A.; Beinart, R.; Boyko, V.; Behar, S.; Matetzky, S., Usefulness of Pretreatment with High-Dose Clopidogrel in Patients Undergoing Primary Angioplasty for St-Elevation Myocardial Infarction. Am J Cardiol 2009, 104 (4), 514-518.
- Gurbel, P. A.; Bliden, K. P.; Zaman, K. A.; Yoho, J. A.; Hayes, K. M.; Tantry, U. S., Clopidogrel Loading with Eptifibatide to Arrest the Reactivity of Platelets: Results of the Clopidogrel Loading with Eptifibatide to Arrest the Reactivity of Platelets (Clear Platelets) Study. Circulation 2005, 111 (9), 1153-1159.
- Hochholzer, W.; Trenk, D.; Frundi, D.; Blanke, P.; Fischer, B.; Andris, K.; Bestehorn, H. P.; Buttner, H. J.; Neumann, F. J., Time Dependence of Platelet Inhibition after a 600-Mg Loading Dose of Clopidogrel in a Large, Unselected Cohort of Candidates for Percutaneous Coronary Intervention. Circulation 2005, 111 (20), 2560-2564.
Reviewer 2 Report
The author presented their retrospective review of the result of internal carotid artery non-branching aneurysms treated by flow diversion therapy alone. Although retrospective and there remains possibility that diagnosis based only on angiographic findings could include small saccular aneurysms, their result support that FDS can be a treatment option for this difficult-to-treat pathology.
I think the presented cases are consecutive, but there are only 4 poor grade cases in their series, as the blood-blister aneurysm often presents with poor neurological condition. And those aneurysms often rebleed soon after presentation, but only 4 patients are presented (or treated) on day 1. So I wonder if there are some selection bias. The authors should mention and clarify that.
Author Response
The author presented their retrospective review of the result of internal carotid artery non-branching aneurysms treated by flow diversion therapy alone. Although retrospective and there remains possibility that diagnosis based only on angiographic findings could include small saccular aneurysms, their result support that FDS can be a treatment option for this difficult-to-treat pathology.
I think the presented cases are consecutive, but there are only 4 poor grade cases in their series, as the blood-blister aneurysm often presents with poor neurological condition. And those aneurysms often rebleed soon after presentation, but only 4 patients are presented (or treated) on day 1. So I wonder if there are some selection bias. The authors should mention and clarify that.
Response: We fully agree with your remarks regarding clinical severity and rebleeding rates seen in blister aneurysms and thank your comment and concern. The cases presented here are consecutive cases of nonbranching internal carotid artery microaneurysms. This group included blister and saccular aneurysm types and we are not aware of any selection bias.
Clinical presentation: note that 4 patients presented with Hunt Hess of 4-5, making 26.6%. We are not aware of any selection bias in this regard as we treat patients with acute SAH as soon as possible. Our policy is to treat patients on admission day. Having said that, SAH day 1 is commonly but not always admission day 1, some patients consult several hours after the ictus, making SAH day and admission days not the same. In addition, some of the patients are transferred from other centers for treatment at our center. Despite these please, note that 13 out of 15 patients were treated on the first 3 days period (10 of them on first 2 days) making this series a group treated at early phase of SAH. Case number 14 was treated on SAH day 4 because that was the day was admitted from another center. Case number 5 was treated on day 8 after an admission angiogram that failed to show aneurysm and repeated angiogram done on day 8 that finally revealed the aneurysm. This second angiogram was followed immediately by treatment. This challenging case was chosen to be shown as Fig 1.
Round 2
Reviewer 1 Report
I'm satisfied with the authors' modifications.